# Novel Targets Regulating the Role of Endothelial Cells and Angiogenesis after Infarction: A RNA Sequencing Analysis

**DOI:** 10.3390/ijms242115698

**Published:** 2023-10-28

**Authors:** María Ortega, Tamara Molina-García, Jose Gavara, Elena de Dios, Nerea Pérez-Solé, Victor Marcos-Garcés, Francisco J. Chorro, Cesar Rios-Navarro, Amparo Ruiz-Sauri, Vicente Bodi

**Affiliations:** 1INCLIVA Biomedical Research Institute, 46010 Valencia, Spain; orcarma@alumni.uv.es (M.O.); tamogar@alumni.uv.es (T.M.-G.); neere_8@hotmail.com (N.P.-S.); vic_mg_cs@hotmail.com (V.M.-G.); francisco.j.chorro@uv.es (F.J.C.); vicente.bodi@uv.es (V.B.); 2Centro de Biomateriales e Ingeniería Tisular, Universidad Politécnica de Valencia, 46010 Valencia, Spain; jose.gavara@outlook.es; 3Centro de Investigación Biomédica en Red (CIBER)-CV, 28029 Madrid, Spain; elenaddll@gmail.com; 4Cardiology Department, Hospital Clínico Universitario, 46010 Valencia, Spain; 5Department of Medicine, University of Valencia, 46010 Valencia, Spain; 6Department of Pathology, University of Valencia, 46010 Valencia, Spain

**Keywords:** myocardial infarction, angiogenesis, endothelial cells, RNA sequencing

## Abstract

Endothelial cells (ECs) are a key target for cardioprotection due to their role in preserving cardiac microvasculature and homeostasis after myocardial infarction (MI). Our goal is to identify the genes involved in post-MI EC proliferation, EC apoptosis, and angiogenesis regulation via RNA-sequencing transcriptomic datasets. Using eight studies from the Gene Expression Omnibus, RNA-sequencing data from 92 mice submitted to different times of coronary ischemia or sham were chosen. Functional enrichment analysis was performed based on gene ontology biological processes (BPs). Apoptosis-related BPs are activated up to day 3 after ischemia onset, whereas endothelial proliferation occurs from day 3 onwards, including an overrepresentation of up to 37 genes. Endothelial apoptosis post-MI is triggered via both the extrinsic and intrinsic signaling pathways, as reflected by the overrepresentation of 13 and 2 specific genes, respectively. BPs implicated in new vessel formation are upregulated soon after ischemia onset, whilst the mechanisms aiming at angiogenesis repression can be detected at day 3. Overall, 51 pro-angiogenic and 29 anti-angiogenic factors displayed altered transcriptomic expression post-MI. This is the first study using RNA sequencing datasets to evaluate the genes participating in post-MI endothelium physiology and angiogenesis regulation. These novel data could lay the groundwork to advance understanding of the implication of ECs after MI.

## 1. Introduction

Myocardial infarction (MI) is the consequence of acute thrombotic occlusion of a coronary artery. Even in patients with successful epicardial artery reperfusion, however, microvascular injury can occur in up to 50% of cases and plays a deleterious role in terms of left ventricular remodeling and patient outcomes [1,2].

Massive disruption in endothelial cell (EC) lining has been reported after MI, which activates cell apoptosis and facilitates capillary obstruction and extravasation of blood content into the interstitium [1,3]. Indeed, severe reduction in microvessel density and abnormal ultrastructure in ECs from small capillaries occur a few minutes after MI induction [4,5]. Contrarily, our own organism quickly releases pro-angiogenic factors tending to repair microcirculation loss, as reflected in both humans and highly controlled models of MI [6,7]. From the very beginning of ischemia, factors with intense pro-angiogenic activity are increased in patients’ plasma [4,8]. Furthermore, the peripheral blood of experiments and patients with MI (from the onset of ischemia until several weeks afterwards) can exert a potent pro-angiogenic stimulus on harvested coronary endothelial cells [4]. This probably mediates the spontaneous regeneration of the microvasculature that, in general, successfully ends a few weeks after reperfusion. Since preliminary results suggesting that complete microvascular restoration after MI might be a realistic endpoint to attenuate left ventricular remodeling and ameliorate prognosis [8,9], unravelling the dynamics of the molecular pathways implicated in the post-MI loss of EC viability and microvascular injury neoangiogenesis-mediated reparation is of great interest to better understand MI pathophysiology and reduce the occurrence of post-MI adverse events.

Over the last few decades, biomedical research has been revolutionized by the appearance of high-throughput sequencing technologies, and several studies have been carried out in the myocardium isolated from experimental models of MI [10,11,12,13,14,15]. A previous meta-analysis from our group using RNA sequencing datasets was focused on identifying the genes involved in the post-MI extracellular matrix turnover, and the specific collagen subunits and metalloproteinases participating in this scenario [16]. However, data specifically addressing EC viability due to the balance between apoptotic and proliferation pathways and key molecular factors regulating post-MI angiogenesis are scarce.

Taken together, the goal of the present study was to carry out a comprehensive quantitative meta-analysis using RNA sequencing transcriptomic datasets to pinpoint dynamic alterations in the mRNA expression of the key genes involved in the biological processes (BPs) that regulate EC proliferation and apoptosis, as well as neoangiogenesis-mediated reparation of microvascular injury after MI.

## 2. Results

### 2.1. Identification of the Selected Studies and Differentially Expressed Genes (DEGs)

We initially identified 15,337 studies using the search term “myocardial infarction”, from which 143 datasets were retrieved from the Gene Expression Omnibus (GEO) and reviewed for eligibility. After removing duplicate series (n = 51), 84 were excluded for the following reasons: knock-out mice (n = 24), single cell or single nuclear (n = 28), evaluation of non-infarcted area (n = 11), mice submitted to pharmacological treatment (n = 14), new-born mice (n = 3), insufficient data about the dataset information (n = 2), and mice submitted to ischemia followed by 4 days of ischemia (n = 2). A total of eight datasets including 92 mice were selected (Table 1). Animals were divided into sham mice (n = 30, without MI induction) and 62 submitted to different coronary ischemia times: 6 h (n = 8), 1 day (n = 16), 3 days (n = 10), 7 days (n = 7), 14 days (n = 10), and 21 days (n = 11). One sample was eventually discarded from the meta-analysis for displaying different behavior from the other included studies in the results from the principal component analysis.

Once DEGs were selected, functional enrichment analysis was then carried out with the gene ontology (GO) BP terms to characterize alterations in the mRNA expression of genes implicated in EC proliferation and apoptosis and angiogenesis regulation.

### 2.2. Scrutinizing Molecular Pathways Regulating EC Viability after MI

Figure 1 displays the upregulated BPs (with an adjusted *p*-value lower than 0.05) related to EC proliferation and apoptosis, identifying bimodal behavior. A clear augmentation in the expression of BPs related to EC apoptosis, namely *the regulation of endothelial cell apoptotic process* (GO: 200035) and *endothelial cell apoptotic process* (GO: 0072577), was noticed at initial stages (days 1–3) of ischemia. Contrarily, significantly increased levels of BPs implicated in endothelium proliferation, specifically *endothelial cell proliferation* (GO: 0001935) and *the regulation of endothelial cell proliferation* (GO: 0001936), were distinguished from day 3 onwards (Figure 1). Overall, the mRNA expression of the genes participating in endothelial apoptosis increased in the first days after ischemia onset, while those involved in EC proliferation exhibited heightened transcriptomic levels from day 3 to chronic phases after MI induction. Subsequently, the overexpressed genes implicated in the regulation of both endothelial proliferation and apoptosis were specifically studied.

### 2.3. Genes Involved in the Regulation of EC Proliferation after MI

Since the increased EC proliferation was initiated at day 3 after coronary occlusion (Figure 1). Table 2 illustrates that 38 genes identified within the BPs of *endothelial cell proliferation (GO: 0001935)* and 36 included in *the regulation of endothelial cell proliferation (GO: 0001936)* exhibited increased mRNA expression. According to our data, *hmox1* and *arg1* displayed the highest mRNA 3 days after coronary occlusion, whereas *mdk*, *scg2*, *gdf2*, and *tnmd* showed strong changes in transcriptomic expression from day 7 to the chronic phases.

Next, we made a specific analysis of genes related to both the positive and negative regulation of this process (Figure 2). We detected an augmented mRNA expression of a total of 37 genes included in the BPs underlying the *regulation of endothelial cell proliferation* (GO: 0001936), which were afterwards overlapped with those from the BPs *Positive regulation of endothelial cell proliferation* (GO: 0001938) and/or *negative regulation of endothelial cell proliferation* (GO: 0001937). The Venn diagram shows altered transcriptomic levels of 18 genes implicated in the positive and 9 in the negative regulation of EC proliferation (Figure 2).

Regarding the positive regulation of EC proliferation, up to five genes (*itgb3*, *vash2*, *bmp2*, *cyba*, and *itga4*) showed heightened transcriptomic expression throughout the entire ischemic process, whereas the mRNA expression of six (*ccl2*, *egr3*, *adora2b*, *apln*, *hmox1*, and *arg1*) was elevated for up to 7 days. A third group (including *scg2*, *htr2b*, *mdk*, *vegfd*, *ccr3*, *il10*, and *hmgb2*) exhibited enlarged transcriptomic levels at days 1–3 post-ischemia induction and remained overrepresented at the chronic phases (Figure 2). In terms of negative regulators of post-MI EC proliferation, although some genes revealed downregulation or even mild upregulation at sub-acute stages, the highest transcriptional levels of up to seven genes (*col4a3*, *sulf1*, *tgfbr1*, *cxcr3*, *apoe*, *sparc*, *thbs1*, *il12b*, and *tnmd*) were detected at day 7 of ischemia, a tendency that persisted at day 21 (Figure 2).

Collectively, EC proliferation after MI started 3 days after coronary occlusion (Figure 2), including overrepresentation of up to 37 genes implicated in both the positive (n = 18) and negative (n = 9) regulation of this process. Genes related to the positive regulation of EC proliferation showed elevated mRNA expression through the entire ischemic period, whereas negative modulation occurred from day 7 onwards (Figure 2).

### 2.4. Apoptosis Signaling Pathway Implicated in EC Apoptosis Post-MI

The BPs involved in EC apoptosis were overrepresented soon after ischemia onset (Figure 1). Table 3 shows a total of 16 genes included in the BPs underlying the *endothelial cell apoptotic process* (GO: 0072577) and *regulation of endothelial cell apoptotic process* (GO: 200035), which displays increased mRNA expression during the initial stages of ischemia (days 1–3). Based on our results, *serpine1* and *thbs1* exhibited the highest transcriptomic changes at very acute phases following MI. Indeed, *tnf*, *angptl4*, *tert*, and *il10* also displayed a significant increase in mRNA levels a few hours after ischemia onset (Table 3).

To pinpoint the contribution of intrinsic and extrinsic apoptotic signaling pathways, all genes included in the BP underlying the *endothelial cell apoptotic process* (GO: 0072577) were selected (n = 55) and overlapped with those from the BPs underlying the *intrinsic apoptotic signaling pathway* (GO: 0097193) (n = 322) and/or *extrinsic apoptotic signaling pathway* (GO: 0097191) (n = 242). The Venn diagram indicated that 14 genes implicated in EC apoptosis belonged to the extrinsic signaling pathway, 2 to the intrinsic, and 3 to both the intrinsic and extrinsic signaling pathways, whereas only 18 displayed higher mRNA expression than sham (Figure 3). The transcriptomic level of the eight genes from the extrinsic pathway was augmented from initial stages post-MI induction to day 21. Of these, *thbs1*, *serpine1*, *scg2,* and *tert* presented a higher Log_2_FoldChange compared to *fasl*, *bmp4*, icam1, and *tnfaip3*. In contrast, the transcriptomic levels of *il4*, *gepr1*, and *foxo3* were downregulated very soon after coronary occlusion (6–24 h) and the gene expression of *fga* remained unaltered (Figure 3). Regarding genes from the intrinsic (*xbp1* and *nfe2l2*) and those included in both the intrinsic and extrinsic pathways (*tnf*, *mapk7*, and *hipk1*), a slight elevation in their mRNA expression was shown up to day 3 after coronary occlusion (Figure 3).

In summary, the genes participating in EC apoptosis are activated up to day 3 after ischemia onset and are triggered via both the extrinsic and intrinsic pathways.

### 2.5. Regulation of Microvasculature Physiology in the MI Setting

Angiogenesis (novel capillaries forming from pre-existing vessels) has been implicated in the resolution of post-MI microvascular injury. In total, 11 BPs related to angiogenesis regulation were overrepresented (Figure 1), all exhibiting upregulation as soon as 6 h after MI induction. According to their dynamics, clear augmentation (with an adjusted *p*-value lower than 0.01) at all times after coronary occlusion was detected in the following angiogenesis-related BPs: *Regulation of angiogenesis* (GO: 0045765) and *Positive regulation of angiogenesis* (GO: 0045766). The involvement of *endothelial cell migration* (GO: 0043542) and *blood vessel endothelial cell migration* (GO: 0043534) BPs was reported within the infarct region at the sub-acute phase (7–14 days). Contrarily, the negative regulation of angiogenesis started 3–7 days post-MI, as reflected by the overexpression of the BP underlying the *negative regulation of angiogenesis* (GO: 0016525), *negative regulation of blood vessel morphogenesis* (GO: 2000181), and *negative regulation of blood vessel diameter* (GO: 0042310) (Figure 1).

This indicates that new vessel formation is activated soon after ischemia onset aiming at restoring microvascular density within the infarcted tissue, whereas the negative regulators of angiogenesis are probably upregulated few days (3 to 7) post-MI. Another mechanism to promote the formation of new vessels into the ischemic heart is the migration of ECs, of which the genes are mainly overrepresented from day 7 onwards. As a next step, an in-depth analysis of those specific molecular targets involved in the positive and negative regulation of angiogenesis was performed.

### 2.6. Molecular Players in Angiogenesis Modulation following MI

As previously shown, the BPs involved in angiogenesis regulation were activated soon after coronary occlusion (Figure 1). Heightened mRNA expression was observed in 111 genes included in the BP underlying the *regulation of angiogenesis* (GO: 0045765), which were subsequently overlapped with those from the BPs underlying the *positive regulation of angiogenesis* (GO: 0045766) and/or *negative regulation of angiogenesis* (GO: 0016525). The Venn diagram shows an overrepresentation of 68 genes implicated in the positive and 29 in the negative regulation of angiogenesis (Figure 4).

Regarding the positive regulation of new vessels formation, mRNA expression peaked at 1 day after MI induction in approximately 20 genes, including *adm2*, *ccl24*, *cxcr2*, and *pgf*, whereas the vast majority showed heightened transcriptomic levels from day 3 to the chronic phase (14–21 days). When dissecting the negative regulators of angiogenesis, 25 genes (i.e., *adamts1*, *adamts9*, *cxcl10*, *tnf*, *fasl*, and *il12*) exhibited augmented mRNA expression at the acute phase (3–7 days post-MI induction) in comparison to the control myocardium (Figure 4).

To sum up, Figure 4 represents the potential candidates to regulate angiogenesis in the MI context. Concretely, the positive regulators of angiogenesis are activated at all ischemic times after coronary occlusion, representing more than a twofold number of genes compared to the negative regulators aiming at promoting the formation of new vessels within an ischemic heart.

### 2.7. Variations in the Transcriptomic Expression of Pro/Anti-Angiogenic Factors

Lastly, we focused specifically on alterations in the mRNA expression of genes encoding pro-angiogenic and anti-angiogenic proteins (Figure 5). In terms of angiogenesis activators (Figure 5A), 51 genes displayed modified transcriptomic levels in the infarcted myocardium of MI animals compared to the controls. Based on their dynamics, seven genes (*csf3*, *cd44*, *itgb3*, *tnf*, *sema3f*, *kit*, and *tgfb1*) showed augmented mRNA levels very soon after coronary occlusion and this trend was sustained at the acute phase (3–7 days). A second group was made up of 18 genes (*ptn*, *fst*, *fn1*, *hgf*, and *ephb2*, among others) displaying elevated mRNA expression from day 7 to the chronic phases; and lastly, 26 genes encoding pro-angiogenic proteins (*ceacam1*, *flt1*, *prox1*, and *fgf1*, among others) showed reduced or slightly augmented levels. Indeed, the genes with the highest mRNA levels were *csf3* (at the initial phase) and *ptn* (at the late phase) (Figure 5A).

We also characterized new vessel formation repressors, detecting 29 genes with altered mRNA expression post-MI (Figure 5B). Of note, *chga* displayed heightened mRNA levels in all MI groups compared to the sham, whereas 11 different genes, including *angpt2*, *angpt4*, *cxcl10*, and *tnfsl15*, were overrepresented from day 7 onwards. Contrariwise, three genes (*vegfb*, *pgk*, and *tie1*) exhibited diminished mRNA expression, and the transcriptomic levels of 13 genes encoding anti-angiogenic proteins (i.e., *timp2*, *angptl1*, *timp3*, and *adamts1*) displayed mild augmentation (Figure 5B).

In summary, the mRNA expression of genes implicated in new vessel formation begins very soon after MI induction, but a high expression throughout the entire process occurs only in the genes implicated in angiogenesis promotion. Nonetheless, molecular pathways, which limit angiogenesis, are also activated from day 3 onwards (Figure 4). We report rearrangements in the mRNA levels of 51 genes encoding pro-angiogenic and 29 anti-angiogenic proteins post-MI (Figure 5).

## 3. Discussion

Using eight independent RNA sequencing studies derived from infarcted tissue isolated at various time points after coronary occlusion, we found that the transcriptomic level of genes participating in EC apoptosis (via the extrinsic and intrinsic pathways) increases up to day 3 after ischemia onset, whereas EC proliferation occurs from day 3 onwards. In terms of angiogenesis modulation, the BPs implicated in new vessel formation are overrepresented as early as ischemia onset, while mechanisms aiming at inhibiting this process are detected from day 3 onwards. Lastly, we identified up to 51 different pro-angiogenic and 29 anti-angiogenic factors displaying altered transcriptomic expression post-infarction (Figure 6).

### 3.1. Changes in the Transcriptomic Expression of the Genes Involved in EC Viability and Proliferation following MI

After ischemic insult, ECs respond in two different ways. Firstly, experimental models revealed abnormal changes in the EC ultrastructure of small capillaries (i.e., thinner, disrupted cytoplasm, chromatin condensation, or absence of transcytotic vesicles) immediately after coronary occlusion [4,5]. Contrariwise, ECs can also withstand prolonged periods of ischemia without a massive loss of cell integrity due to the activation of potential protective mechanisms [17]. The dual behavior displayed by ECs post-MI highlights the need for an in-depth characterization of rearrangements in the expression of key genes participating in post-MI EC proliferation.

Based on our results, the transcriptomic expression of the genes implicated in endothelial apoptosis is boosted soon after ischemia onset. Conversely, those involved in EC proliferation showed increased mRNA levels from day 3 onwards, as reflected by the augmentation of 37 different genes, 18 of them being the positive and 9 the negative regulators of EC proliferation (Figure 2). Similarly, a recent study performed via single-cell RNA sequencing in ECs isolated post-MI concluded that positive modulation of EC proliferation peaks at days 1 to 5 after ischemic insult [18]. Both studies derived from next-generation sequencing approaches are in line with the dynamics of microvascular perfusion reported at bedside. In a series of patients with reperfused ST-segment elevation MI (STEMI) undergoing sequential cardiovascular magnetic resonance studies, 44% displayed significant microvascular injury 1 week after MI, and most exhibited complete recovery at 6 months [7].

According to Table 2, the mRNA levels of up to 38 genes involved in the regulation of EC proliferation were significantly enlarged. Concretely, *hmox1* and *arg1* displayed the highest transcriptomic expression 3 days after coronary occlusion, whereas *mdk*, *scg2*, *gdf2*, and *tnmd* showed strong changes in mRNA levels from day 7 onwards. In the MI scenario, heme oxygenase (*hmox1*) has been demonstrated to reduce infarct size and left ventricular remodeling in murine models of permanent ischemia [19], and exert cardioprotective effects via attenuating cardiomyocyte senescence [20], whereas midkine (*mdk*) administration prevents left ventricular remodeling by promoting angiogenesis and reducing collagen deposition and apoptosis activation [21]. Indeed, the loss of growth differentiation factor 2 (*gdf2*) promotes cardiac fibrosis, collagen degradation, and cardiac rupture after MI via metalloproteinase 9 [22]. Lastly, circulating levels of arginase 1 (*arg1*) are augmented in MI patients compared to the controls, and positively correlate with the gensini score [23]. Contrariwise, from our knowledge, studies regarding the participation of secretogranin II (*scg2*) and tenomodulin (*tnmd*) in an MI scenario are scarce.

Vascular endothelial growth factor actively participates in EC proliferation to spontaneously promote post-MI microvascular injury resolution via angiogenesis activation [8]. Our data revealed the overrepresentation of several positive regulators of EC proliferation (i.e., *scg2*, *ccr3*, and *vash2*) (Figure 2), which has been linked to the vascular endothelial growth factor signaling pathway in other disease entities [24,25]. This highlights the utility of further investigating their specific involvement in the pathophysiology of MI as promising biomarkers and/or novel therapeutic options.

### 3.2. Dissecting EC Apoptosis after MI

As mentioned above, EC apoptosis triggered by ischemia seems to play a pivotal role in the loss of post-MI microvascular integrity. Indeed, in vitro and in vivo approaches carried out in the MI setting consistently demonstrate that ECs undergo apoptosis, a programmed cell death mechanism characterized by cell shrinkage and chromatin condensation. For instance, in a cohort of STEMI patients, serum isolated 24 h after coronary revascularization induced apoptosis in ECs in vitro; moreover, the magnitude of this serum-induced the loss of EC viability paralleled the extent of cardiovascular magnetic resonance-derived edema, intramyocardial hemorrhage and microvascular obstruction [26]. Another investigation of 153 STEMI patients found an enlarged number of EC-derived microvesicles related to apoptosis [27]. Similarly, our study illustrates the overrepresentation of EC-apoptosis-related BPs at initial stages (day 1–3) after ischemia onset, including up to 55 upregulated genes involved in EC apoptosis (i.e., *tnf*, *mapk7*, and *fasl*, among others) (Figure 3).

Furthermore, we also shed light on the specific apoptotic pathway (extrinsic or extrinsic) activated on ECs post-MI. Expanding on a previous study performed by single-cell RNA sequencing in ECs submitted to permanent myocardial ischemia demonstrated the implication of the intrinsic signaling pathway via *bax* and *trp53* upregulation [18], our data showed heightened mRNA expression of the genes involved in intrinsic (n = 2) and extrinsic (n = 13) signaling pathways (Figure 3).

The extrinsic apoptotic pathway is triggered once ligands (i.e., tumor necrosis factor receptor and Fas) interact with their corresponding death receptors on the cell surface [28]. In this scenario, *tnfaip3*, strongly activated by tumor necrosis factor and interleukin-1 [29], displays higher transcriptomic levels on the infarcted myocardium throughout the entire ischemic process compared to the control myocardium. A recent study reported a positive correlation between the TNFAIP3 protein concentration in the circulating monocytes from MI patients and high-sensitivity troponin T levels [30].

Indeed, *thbs1*, *serpine1*, *scg2*, and *tert* regulate the extrinsic apoptotic pathway and present higher mRNA levels from day 1 onwards. Specifically, thrombospondin 1 (*thbs1*) participates in the MI context via an inhibition of inflammation by promoting neutrophil apoptosis, myofibroblast differentiation, the acceleration of fibrotic scar formation, and activation of apoptosis in ECs at the microvascular level [31], whereas telomerase reverse transcriptase (*tert*) overrepresentation correlates with reduced infarct size and better systolic function in murine models of permanent ischemia [32], and a decay in cardiomyocyte apoptosis [33]. Regarding *serpine 1*, increased circulating levels are associated with a higher 5-year risk of death in STEMI [34] and ventricular disfunction [35]. In experimental models, the presence of serpine 1 is probably increased via transforming growth factor-β and tumor necrosis factor-α, and participates in cardiac fibrosis and ventricular remodeling [36].

The intrinsic pathway, meanwhile, is triggered via the endoplasmic reticulum and mitochondria [28]. Our findings point to an overrepresentation of *xbp1*, a transcription factor that modulates not only the cellular response to endoplasmic reticulum stress, but also vascular endothelial growth factor-induced angiogenesis in adult tissues submitted to ischemia [37]. Despite its role in the mechanisms underlying various cardiovascular diseases, including cardiac hypertrophy and heart failure [37], few studies have addressed the involvement of *xbp1* in the MI context.

Collectedly, our study reveals new insights into molecular targets probably implicated in EC apoptosis regulation, but that have barely been scrutinized in the MI context so far. These novel data could establish the basis for designing further experiments aiming at broadening the understanding of EC participation post-MI, as well as setting up new potential circulating biomarkers of compromised resultant cardiac structure or therapeutic options to downregulate EC apoptosis after ischemia/reperfusion injury.

### 3.3. In-Depth Characterization of Genetic Targets Modulating Post-MI Angiogenesis

Experimental and cardiac-imaging-based studies have robustly demonstrated the natural endogenous tendency to complete myocardial microvascular injury repair post-MI [4,7]. This occurs in the vast majority of reperfused STEMI patients and is associated with salutary effects in terms of reverse left ventricular remodeling [1,2,7]. However, we recently reported that a certain degree of microvascular damage (as derived from cardiovascular magnetic resonance) can persist in chronic phase after STEMI in a small subset of cases [7]. This delay in microvascular repair exerts deleterious effects in terms of adverse left ventricular remodeling. Therefore, attempts to better understand the molecular mechanisms underlying microvascular injury repair could be pivotal for advancing toward strategies aimed at preservation, and if necessary, controlled repair of microvasculature after MI. This could ultimately inspire future strategies aimed at minimizing structural damage and improving patient survivorship after MI.

Angiogenesis consists of the development of newly formed micro-vessels from pre-existing capillaries in response to different stimuli, including ischemia [4,6,8]. According to our results, up to 30 genes implicated in the positive regulation of angiogenesis displayed massive overrepresentation as soon as 6 h following MI induction, thus indicating the rapid initiation of this process after coronary occlusion. Globally, we reported an overrepresentation of 68 genes encoding proteins implicated in the positive regulation of angiogenesis, most peaking at the sub-acute (3–7 days) phase post-MI (Figure 4). Another study via single-cell transcriptome analysis concluded that ten different heterogeneous ECs states are identified after ischemia insult, exhibiting different roles in terms of angiogenesis promotion [38]. Indeed, in parallel to the myriad pro-angiogenic stimuli, our investigation also reveals the overrepresentation of almost 30 genes aimed at avoiding an excessive angiogenic process.

Controlled, orchestrated modulation of the balance between pro-/anti-angiogenic factors has been effective in minimizing infarct size and promoting new capillary formation in experimental MI models [6,8]. Overall, further understanding angiogenesis regulation after myocardial ischemia is crucial to suggest novel targets aiming at promoting microvascular restoration.

The present study makes use of recent methodological advances in mass sequencing and systematic recording within the scientific community of the results obtained from studies with bulk RNA sequencing analysis from the infarcted myocardium in highly controlled mouse models [10,11,12,13,14,15]. According to our novel data, alterations in the transcriptomic levels of 51 genes encoding pro-angiogenic and 29 anti-angiogenic proteins occur in the infarcted myocardium (Figure 5). Of these, the pleiotrophin gene (*ptn)*, encoding a cytokine with vessel formation in neonatal hearts, displayed exponentially increased mRNA expression in our meta-analysis. In the field of MI, the circulating concentration of pleiotrophin was independently associated with acute coronary syndrome [39] and has also been implicated in cardiomyocyte apoptosis [39], yet few studies have evaluated its efficacy in terms of angiogenesis modulation after coronary occlusion.

Finally, it should be borne in mind that augmented gene expression does not always translate to the protein level. For that reason, future research should aim to corroborate these results at the protein level, both within the experimental models of myocardial ischemia and, more importantly, clinical scenario. However, our study proposes several key factors, which potentially participates in MI pathophysiology and could help establish the basis for designing further clinical and experimental studies addressed at broadening the range of potential angiogenesis regulators participating in post-MI microvascular repair.

### 3.4. Limitations of the Study

In our study, we are unable to detect DEGs among different datasets due to the statistical analysis performed.

These results were obtained at the gene level and in mice submitted to permanent coronary ischemia; therefore, further validation at the protein level and in samples from MI patients is necessary.

## 4. Materials and Methods

For the meta-analysis, we selected studies with datasets including the search term “myocardial infarction” in the GEO database [40] up to September 2021. Inclusion criteria were studies performed in *Mus musculus*, animals submitted to the MI or sham, and studies with bulk RNA sequencing analysis from the infarcted myocardium. Exclusion criteria were studies carried out using single-cell or single-nucleus RNA sequencing, mice submitted to any pharmacological or mechanical intervention, studies with newborn mice, analysis carried out using the blood or non-infarcted myocardium, and those not correctly filtered by GEO’s own filters.

The datasets were evaluated following inclusion and exclusion criteria as part of the complete study flow chart detailed in the PRISMA Flow Diagram. Afterward, the SRA Run Selector was used to obtain the accessions and download the sequence data files using the SRA Toolkit. The alignment of the FASTA files to the reference genome (GRCm38) was carried out with Bowtie2 [41]. By using HTSeq [42], the read counts for each gene were calculated to obtain the expression levels.

We conducted the following computational analyses in the R environment [43]. The read counts for each study were normalized (except for the GSE83350 dataset which only has one animal per study group), saved, and combined with the normalized counts of all the studies in a unique DESeqDataSet. The design also considered the batch effect (of the study from which the data have been obtained) to remove it. Differential expression analysis was performed with DESeq2 (1.30.1) [44] to compare all the infarcted groups with the controls. A *p*-adjusted value < 0.05 and a Log_2_FoldChange ≥ 2 cutoff were used for determining DEGs. Briefly, DESeq2 normalizes and calculates the dispersion of the raw readings of each study group and carries out and performs a hypothesis using a Wald test.

Functional enrichment analysis of the DEGs was performed with the Cluster Profiler package [45] using GO annotations [46] based on the February 2023 data grouping of the DEGs according to the BP category and adjusting the *p*-value with the Benjamin–Hochberg method. Using ClusterProfiler, the comparison of the BPs between different study groups was carried out (between mice with different ischemia times and control), obtaining the functional profiles of each study group.

The BPs with a *p*-value less than 0.05 were selected. As our objective was to evaluate the BPs related to angiogenesis and apoptosis, the descriptions were filtered using the terms “angiogenesis”, “blood vessel”, “endothelial”, and “apoptotic”. Graphical representations were carried out with the ggplot2 (3.3.5) [47], pheatmap (1.0.1) [48], and eulerr (7.0.0) packages [49].

## 5. Conclusions

To our knowledge, this is the first study employing RNA sequencing datasets to gain insight into EC proliferation and apoptosis and angiogenesis regulation in the MI setting. As derived from this novel approach, the genetic upregulation of the molecular mechanisms directed at promoting endothelial apoptosis (via the extrinsic and intrinsic signaling pathways) increases up to day 3 after MI induction, as reflected by the overrepresentation of up to 55 genes, while endothelial proliferation takes place from day 3 onwards including the overrepresentation of up to 37 genes. The activation of angiogenesis regulation occurs soon after coronary occlusion, and post-MI, up to 51 different pro-angiogenic and 29 anti-angiogenic factors are overexpressed. These results could set the foundations for designing further experimental and clinical studies focused on promoting new vessel formation in the setting of MI.

## Figures and Tables

**Figure 1 ijms-24-15698-f001:**
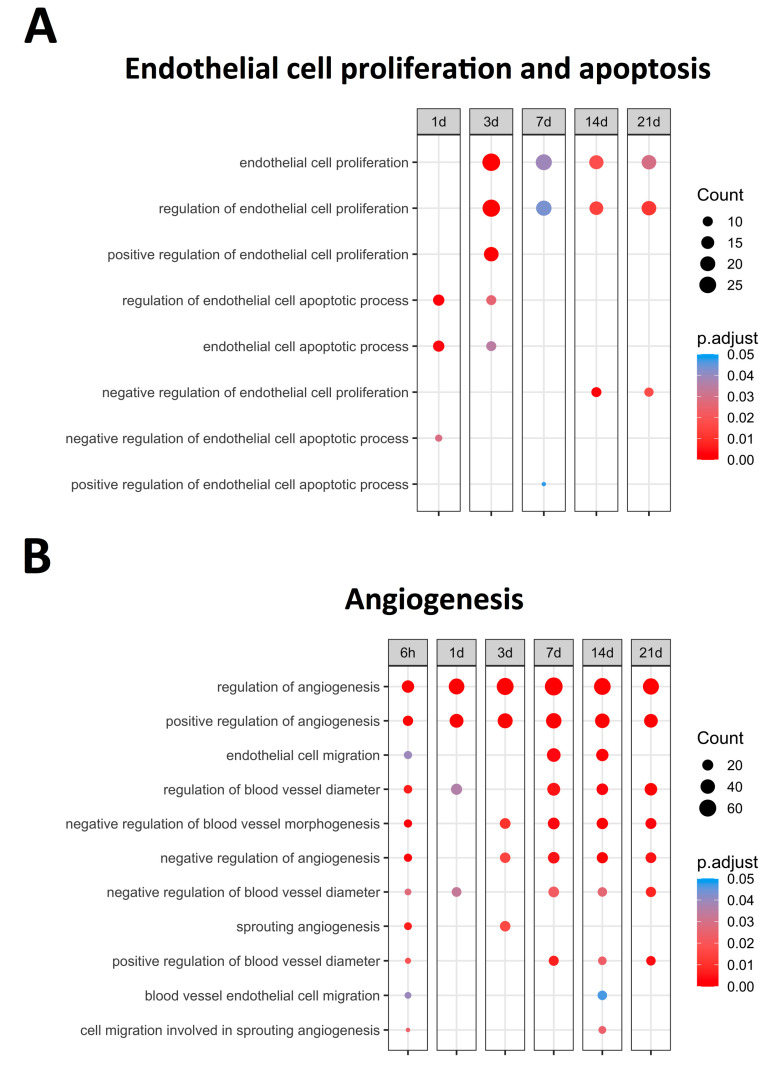
Endothelial cell proliferation, apoptosis-related BPs, (**A**) and angiogenesis-related BPs (**B**) overrepresented at each time point after coronary occlusion. Myocardial infarction groups are in the columns, while the rows represent the different BPs. The size of the circle indicates the number of overexpressed genes, whereas the colors ranging from red to blue represent adjusted *p*-value, as shown in the legend. The red circles in the figure indicate lower *p*-values, while the blue represent higher *p*-values. Abbreviation. BP: biological process.

**Figure 2 ijms-24-15698-f002:**
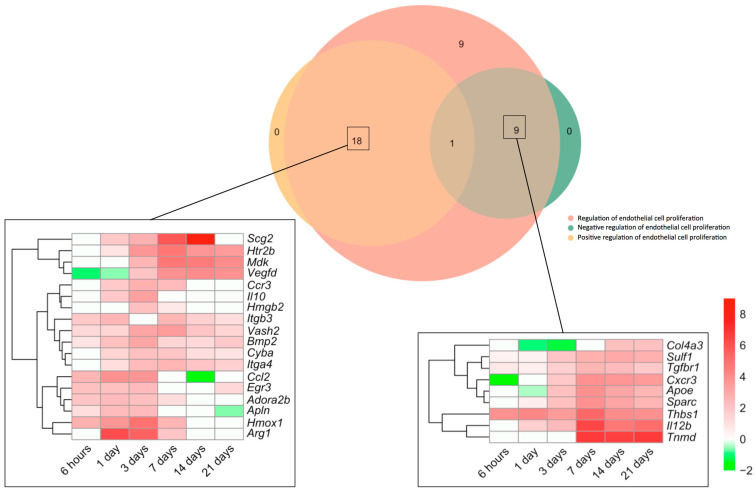
Genes involved in endothelial cell proliferation regulation in the context of myocardial infarction. The Venn diagram represents GO included in the BPs underlying the *regulation of endothelial cell proliferation* (GO: 0001936), *positive regulation of endothelial cell proliferation* (GO: 0001938), and *negative regulation of endothelial cell proliferation* (GO: 0001937). According to the Venn diagram, 18 genes were implicated in the positive and 9 in the negative regulation of angiogenesis. Heatmaps of the Log2-fold change of the genes included in the cross-section groups are represented at each time point after ischemia onset. Myocardial infarction groups are in the columns and genes in the rows. Gene intensities were log2-transformed and are displayed in colors ranging from red to green, as shown in the key. The red squares in the figure represent high-expression genes, while green squares represent low-expression genes. The rows are clustered using correlation distance and average linkage. Abbreviations. BP: biological process. GO: gene ontology.

**Figure 3 ijms-24-15698-f003:**
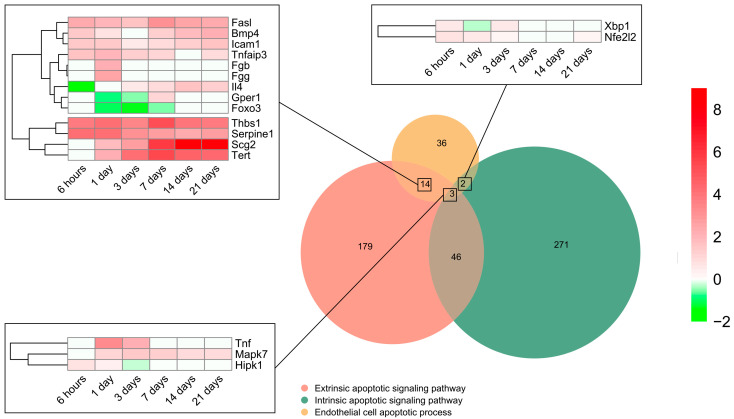
Endothelial cell apoptosis and extrinsic and intrinsic apoptotic signaling pathway-related BPs differentially expressed at each time point after myocardial infarction. The Venn diagram represents the GO included in the BPs underlying the *endothelial cell apoptotic process* (GO: 0072577), *intrinsic apoptotic signaling pathway* (GO: 0097193), and *extrinsic apoptotic signaling pathway* (GO: 0097191). According to the Venn diagram, 14 genes implicated in endothelial cell apoptosis belong to the extrinsic signaling pathway, 2 to the intrinsic, and 3 to both the intrinsic and extrinsic signaling pathways. Heatmaps of the log2-fold change of the genes included in the cross-section groups are represented at each time point after ischemia onset. Myocardial infarction groups are in the columns and genes in the rows. Gene intensities were log2-transformed and are displayed in colors ranging from red to green, as shown in the key. The red squares in the figure represent high-expression genes, while the green squares represent low-expression genes. The rows are clustered using correlation distance and average linkage. Abbreviations. BP: biological process. GO: gene ontology.

**Figure 4 ijms-24-15698-f004:**
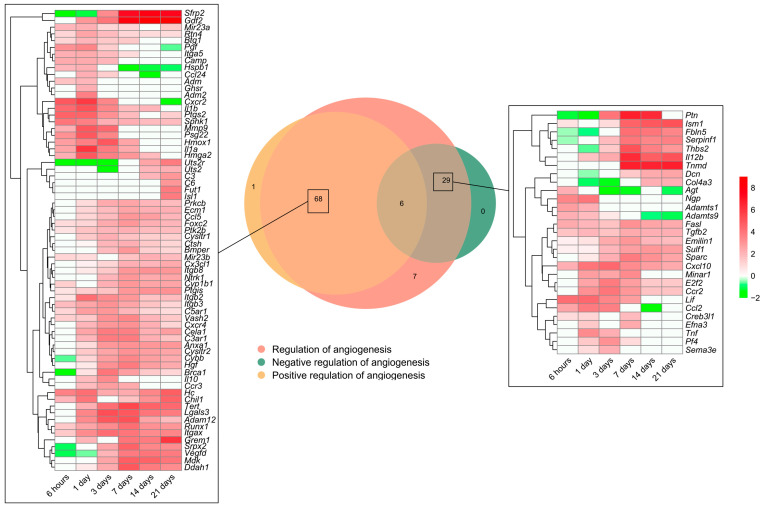
Genes related to angiogenesis regulation in a myocardial infarction scenario. The Venn diagram represents the GO included in the BPs underlying the *regulation of angiogenesis* (GO: 0045765), *positive regulation of angiogenesis* (GO: 0045766), and *negative regulation of angiogenesis* (GO: 0016525). According to the Venn diagram, 68 genes were implicated in the positive and 29 in the negative regulation of angiogenesis. Heatmaps of the log2-fold change of the genes included in the cross-section groups are represented at each time point after ischemia onset. Myocardial infarction groups are in the columns and genes in the rows. Gene intensities were log2-transformed and are displayed in colors ranging from red to green, as shown in the key. The red squares in the figure represent high-expression genes, while the green squares represent low-expression genes. The rows are clustered using correlation distance and average linkage. Abbreviations. BP: biological process. GO: gene ontology.

**Figure 5 ijms-24-15698-f005:**
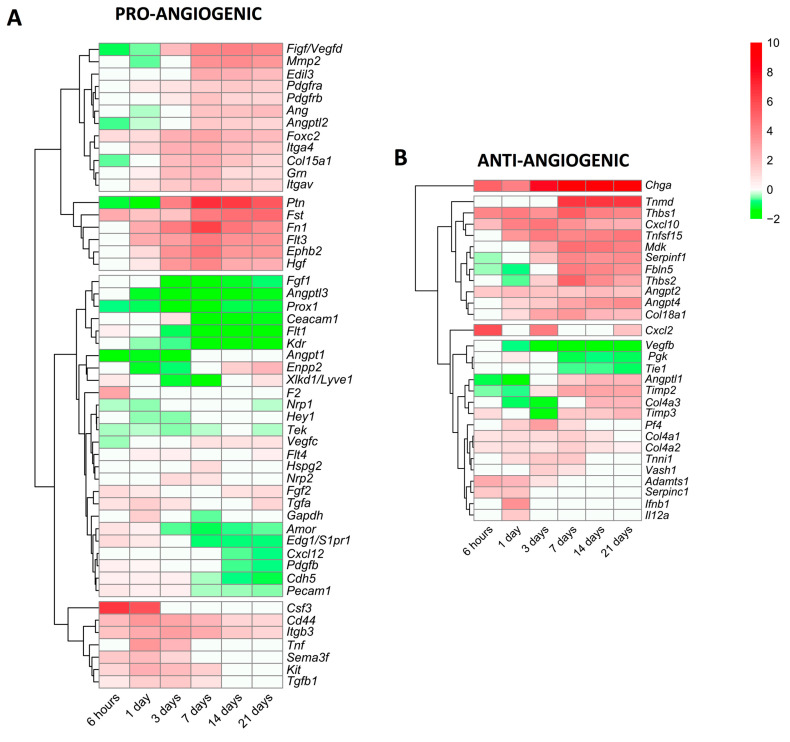
Heatmap of the log2-fold change of pro-angiogenic (**A**) and anti-angiogenic (**B**) genes at different time points after coronary reperfusion. Columns show myocardial infarction groups and the different genes encoding angiogenesis-related genes are in the rows. Gene intensities were log2-transformed and are displayed as colors ranging from red to green, as shown in the key. The red squares in the figure represent high-expression genes, while the green squares represent low-expression genes. Both the rows and columns are clustered using correlation distance and average linkage.

**Figure 6 ijms-24-15698-f006:**
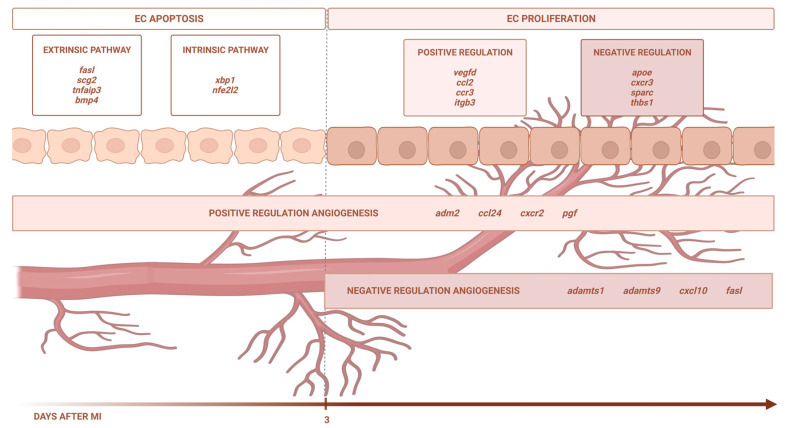
Central illustration. The transcriptomic level of the genes participating in EC apoptosis (via the extrinsic and intrinsic pathways) increases up to day 3 after ischemia onset, whereas EC proliferation occurs from day 3 onwards. In terms of angiogenesis modulation, the BPs implicated in new vessel formation are overrepresented as early as ischemia onset, while the mechanisms aiming at inhibiting this process are detected from day 3 onwards. Image created with BioRender.com, accessed on 13 September 2023.

**Table 1 ijms-24-15698-t001:** Summary of GEO datasets for meta-analysis.

Dataset	Study	Number of Animals
GSE153494	[10]	12
GSE151834	[11]	20
GSE114695	[12]	18
GSE153485	[13]	20
GSE154072	N/A	3
GSE153493	N/A	9
GSE104187	[14]	8
GSE83350	[15]	2
**Total**		92

Abbreviation. N/A: not available.

**Table 2 ijms-24-15698-t002:** Summary of the main genes related to EC proliferation and the regulation of EC proliferation.

		Gene	3 Days of Ischemia		7 Days of Ischemia		14 Days of Ischemia		21 Days of Ischemia	
			Log2-Fold Change	Adjusted *p*-Value	Log2-Fold Change	Adjusted *p*-Value	Log2-Fold Change	Adjusted *p*-Value	Log2-Fold Change	Adjusted *p*-Value
Regulation of endothelial cell proliferation (GO: 0001936)	Endothelial cell proliferation (GO: 0001935)	*apoe*	2	1.37 × 10^−35^	4	5.03 × 10^−71^	3	2.94 × 10^−25^	3	2.06 × 10^−36^
		*il12b*	2	0.001	6	5.77 × 10^−15^	5	1.87 × 10^−5^	5	5.55 × 10^−10^
		*hmox1*	5	1.65 × 10^−35^	3	5.49 × 10^−8^	-	-	-	-
		*cyba*	2	1.64 × 10^−35^	2	1.37 × 10^−21^	1	0.0002	0.92	3.96
		*aldh1a2*	3	7.91 × 10^−19^	1	0.0049	-	-	-	-
		*il10*	3	8.69 × 10^−7^	-	-	-	-	-	-
		*sulf1*	2	3.29 × 10^−66^	3	3.49 × 10^−85^	3	8.70 × 10^−51^	3	4.86 × 10^−88^
		*adora2b*	2	1.27 × 10^−15^	1	0.0002	-	-	-	-
		*sparc*	2	7.14 × 10^−46^	4	1.42 × 10^−95^	3	4.03 × 10^−35^	3	7.79 × 10^−45^
		*arg1*	5	2.22 × 10^−15^	2	0.02	-	-	-	-
		*itgb3*	3	2.90 × 10^−46^	3	4.44 × 10^−24^	2	1.25 × 10^−5^	1	4.72 × 10^−6^
		*tnf*	2.17	3.22 × 10^−8^	-	-	-	-	-	-
		*htr2b*	4	8.53 × 10^−30^	5	1.58 × 10^−38^	4	3.92 × 10^−13^	3	1.17 × 10^−21^
		*itga4*	2	1.93 × 10^−30^	3	3.91 × 10^−26^	2	2.47 × 10^−8^	2	6.64 × 10^−12^
		*mdk*	3	1.09 × 10^−33^	5	2.93 × 10^−82^	5	3.76 × 10^−44^	4	7.04 × 10^−67^
		*bmp2*	3	1.70 × 10^−41^	2	2.63 × 10^−8^	1	0.001	2	1.38 × 10^−11^
		*vegfd*	2	1.63 × 10^−21^	4	1.46 × 10^−57^	4	4.99 × 10^−33^	4	4.85 × 10^−60^
		*egr3*	2	6.23 × 10^−10^	-	-	-	-	1	0.003
		*ccl12*	3	6.80 × 10^−20^	1	7.22 × 10^−5^	-	-	1	0.0007
		*ccl2*	4	9.61 × 10^−26^	-	-	−1	0.02	-	-
		*ccr3*	3	1.77 × 10^−6^	2	0.0003	-	-	-	-
		*apln*	2	9.33 × 10^−24^	-	-	-	-	−0.627	0.02
		*vash2*	3	2.15 × 10^−22^	4	9.39 × 10^−22^	2	0.0002	1	0.0002
		*thbs1*	3	2.88 × 10^−24^	5	4.94 × 10^−39^	4	5.43 × 10^−13^	4	2.84 × 10^−23^
		*scg2*	3	0.0002	6	1.38 × 10^−13^	9	1.76 × 10^−17^	9	4.23 × 10^−30^
		*hmgb2*	2	2.84 × 10^−23^	0.765	0.008	-	-	-	-
		*gdf2*	4	5.39 × 10^−4^	8	2.25 × 10^−9^	9	1.42 × 10^−7^	10	4.36 × 10^−15^
		*tgfbr1*	2	1.99 × 10^−32^	2	4.62 × 10^−50^	2	1.97 × 10^−23^	2	1.52 × 10^−28^
		*igf1*	2	2.04 × 10^−20^	4	1.95 × 10^−51^	3	3.19 × 10^−19^	2	4.35 × 10^−24^
		*thbs4*	2	2.24 × 10^−14^	4	3.03 × 10^−47^	4	6.06 × 10^−24^	4	1.56 × 10^−50^
		*ecm1*	2	1.60 × 10^−26^	3	2.69 × 10^−34^	2	2.47 × 10^−15^	2	5.88 × 10^−27^
		*tnmd*	-	-	7	7.98 × 10^−15^	7	8.13 × 10^−9^	7	4.29 × 10^−15^
		*cxcr3*	2	1.22 × 10^−6^	4	5.03 × 10^−21^	4	4.30 × 10^−11^	3	1.53 × 10^−17^
		*col4a3*	−1	2.36 × 10^−6^	-	-	−4	1.17 × 10^−8^	2	6.80
		*lep*	-	-	-	-	-	-	7	0.0007
		*bmp4*	-	-	1	2.79 × 10^−7^	2	1.15 × 10^−8^	2	5.83 × 10^−22^
		*loxl2*	3	3.29 × 10^−66^	4	5.20 × 10^−67^	2	2.43 × 10^−15^	2	4.03 × 10^−19^
		*bmper*	2	2.80 × 10^−19^	3	2.95 × 10^−37^	1	0.0004	1	3.00 × 10^−17^

Abbreviation. EC: endothelial cell.

**Table 3 ijms-24-15698-t003:** Summary of the main genes related to EC apoptosis and regulation of EC apoptosis.

		Gene	1 Day of Ischemia		3 Days of Ischemia	
			Log2-Fold Change	Adjusted *p*-Value	Log2-Fold Change	Adjusted *p*-Value
Endothelial cell apoptotic process (GO: 0072577)	Regulation of endothelial cell apoptotic process (GO: 200035)	*fasl*	2.07	0.0001	1.55	0.01
		*angptl4*	3.28	1.62 × 10^−23^	2.78	1.45 × 10^−11^
		*tnfaip3*	2.06	1.75 × 10^−22^	1.26	1.95 × 10^−6^
		*tert*	2.26	3.77 × 10^−6^	4.18	1.67 × 10^−17^
		*tnf*	3.24	3.18 × 10^−23^	2.17	3.22 × 10^−8^
		*fgf21*	2.59	0.005	-	-
		*cd40lg*	2.84	0.005	-	-
		*fgb*	2.22	0.02	-	-
		*fgg*	2.53	0.02	-	-
		*ccl12*	2.04	5.07 × 10^−17^	2.70	6.79 × 10^−20^
		*serpine1*	4.23	3.35 × 10^−78^	3.03	6.65 × 10^−27^
		*thbs1*	4.17	2.62 × 10^−52^	3.47	2.88 × 10^−24^
		*col18a1*	1.07	9.53 × 10^−15^	2.88	1.80 × 10^−67^
		*il10*	1.81	0.003	3.22	8.68 × 10^−7^
		*itga4*	1.14	7.81 × 10^−11^	2.38	1.93 × 10^−30^
		*scg2*	1.87	0.006	2.73	0.0001

Abbreviation. EC: endothelial cell.

## Data Availability

Not applicable.

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
