# Peer review of "Novel Targets Regulating the Role of Endothelial Cells and Angiogenesis after Infarction: A RNA Sequencing Analysis"

_ijms, 2023, doi:10.3390/ijms242115698_

Round 1
Reviewer 1 Report
Comments and Suggestions for Authors
This study by Ortega M. et al, entitled "Novel targets regulating endothelial cell action and angiogenesis after myocardial infarction: an RNA sequencing analysis," aimed to identify genes involved in the regulation of endothelial cell proliferation, apoptosis, and angiogenesis after myocardial infarction by using the RNA sequencing transcriptome database of eight studies in the Gene Expression Omnibus (GEO). Using functional enrichment analysis, the authors found that biological processes associated with apoptosis (involving 14 extrinsic and 2 intrinsic genes) were activated until day 3. In contrast, endothelial proliferation (involving 51 pro-angiogenic and 29 anti-angiogenic factors) occurred from day 3 onward. Using bioinformatics methods, the authors clearly explained the role of endothelial cells between apoptosis and proliferation after myocardial infarction at different time points. Notably, this team published Meta-analysis of Extracellular Matrix Dynamics after Myocardial Infarction Using RNA Sequence Transcriptome Database (PMID: 36555255) utilizing the same 8 GEO datasets. The authors should include references to previous findings, discussions, and citations in this manuscript. I recommend that the authors include and discuss previous research findings in this manuscript. The authors should revise Figure 1 and Table 1 which were identical or similar to the previously published article (Int J Mol Sci 2022 Dec 9;23(24):15615). Additionally, the authors can make all the code that analyzes the data and reproduces the results in this manuscript available on GitHub.
Author Response
Reviewer #1: This study by Ortega M. et al, entitled "Novel targets regulating endothelial cell action and angiogenesis after myocardial infarction: an RNA sequencing analysis," aimed to identify genes involved in the regulation of endothelial cell proliferation, apoptosis, and angiogenesis after myocardial infarction by using the RNA sequencing transcriptome database of eight studies in the Gene Expression Omnibus (GEO). Using functional enrichment analysis, the authors found that biological processes associated with apoptosis (involving 14 extrinsic and 2 intrinsic genes) were activated until day 3. In contrast, endothelial proliferation (involving 51 pro-angiogenic and 29 anti-angiogenic factors) occurred from day 3 onward. Using bioinformatics methods, the authors clearly explained the role of endothelial cells between apoptosis and proliferation after myocardial infarction at different time points.
Notably, this team published Meta-analysis of Extracellular Matrix Dynamics after Myocardial Infarction Using RNA Sequence Transcriptome Database (PMID: 36555255) utilizing the same 8 GEO datasets. The authors should include references to previous findings, discussions, and citations in this manuscript. I recommend that the authors include and discuss previous research findings in this manuscript.
First, we would like to thank the reviewer for the suggestions. It is right that our group has recently published a meta-analysis was performed aiming at further understanding the interstitial changes following myocardial infarction. Briefly, in this previous article, biological processess implicated in response to extracellular stimulus, regulation of extracellular matrix organization, and extracellular matrix disassembly were detected soon after ischemia onset. Extracellular matrix disassembly occurred between days 1 to 7 post-MI, compared with extracellular matrix assembly from day 7 onwards. We were able to identify altered mRNA expression of 19 matrix metalloproteinases and 4 tissue inhibitor of metalloproteinases at post-infarcted interstitium remodeling as well as altered transcriptomic expression of 42 genes encoding 26 collagen subunits at fibrotic stage. To our knowledge, this was the first meta-analysis using RNA-sequencing datasets to evaluate post-infarcted cardiac interstitium healing, revealing previously unknown mechanisms and molecules actively implicated in extracellular matrix remodeling post-infarction.
This issue has been raised at different sections of the revised manuscript:
Introduction section, page 2, paragraph 2:
“A previous meta-analysis from our group using RNA-sequencing datasets was focused on identifying genes involved in post-MI extracellular matrix turnover and the specific collagen subunits and metalloproteinases participating in this scenario [16]. However, data specifically addressing EC viability due to the bal-ance between apoptotic and proliferation pathways and key molecular factors regulating post-MI angiogenesis are scarce”.
Reference:
- Ortega, M.; Ríos-Navarro, C.; Gavara, J.; de Dios, E.; Perez-Solé, N.; Marcos-Garcés, V.; Ferrández-Izquierdo, A.; Bodí, V.; Ruiz-Saurí, A. Meta-Analysis of Extracellular Matrix Dynamics after Myocardial Infarction Using RNA-Sequencing Transcriptomic Database. Int J Mol Sci 2022, 23, 15615.
The authors should revise Figure 1 and Table 1 which were identical or similar to the previously published article (Int J Mol Sci 2022 Dec 9;23(24):15615).
We agree with this comment. For that reason, in the revised version of the manuscript, Table 1 was removed, and Table 1 was changed according to previously published article (Ortega M, et al. 2022).
Additionally, the authors can make all the code that analyzes the data and reproduces the results in this manuscript available on GitHub.
The reviewer is right. It would be interesting to upload the code on GitHub. However, we are still making some additional analysis using this code since we would also like to scrutinize changes regarding non-coding RNA. For that reason, as soon as we published the third article from this RNA-sequencing datasets, we will upload all the code on GitHub.
Reviewer 2 Report
Comments and Suggestions for Authors
The authors aimed in this study to identify genes involved in post-myocardial infarction (MI) endothelial cell (EC) functions using RNA-sequencing data from mice subjected to coronary ischemia. The results show that apoptosis-related processes are active up to day 3 after ischemia, while EC proliferation begins from day 3 onwards. Apoptosis in ECs post-MI involves both extrinsic and intrinsic pathways, and there are changes in gene expression related to new vessel formation and angiogenesis regulation. This study provides valuable insights into the role of ECs after MI, potentially advancing our understanding of their significance. My questions and comments are;
1. The authors repeated the information on microvascular injury and the importance of understanding its molecular pathways in the introduction. A more concise and direct presentation could make an impact on the introduction.
2. There are missing specific research gaps to explain the significance of microvascular injury and neo-angiogenesis regarding the study aims. Elaboration is required to address the hypothesis /objectives such as the specific genes or pathways the study intends to investigate.
3. Could authors elaborate on the process of selecting differentially expressed genes (DEGs) and the methodology used for functional enrichment analysis with Gene Ontology terms? The methods sections need improvement. What statistical analysis was used for each experiment?
4. Were there any significant findings or trends observed in the functional enrichment analysis related to EC proliferation, apoptosis, and angiogenesis regulation?
5. Could the author elaborate on the significance of the bimodal behavior observed in the upregulated biological processes (BPs) related to endothelial cell (EC) proliferation and apoptosis after MI?
6. Results are not well explained. What specific genes or factors were identified within the Regulation of endothelial cell apoptotic process and Endothelial cell apoptotic process biological processes during the initial stages of ischemia (days 1-3)?
7. The authors should provide more details about the genes or pathways that were associated with Endothelial cell proliferation and Regulation of endothelial cell proliferation from day 3 onwards in the results.
8. Interested to know whether any specific genes or pathways exhibited particularly strong changes in expression within these biological processes during the study.
9. While the result section 2.3 reports changes in gene expression patterns, it doesn't delve into the potential functional implications of these changes. Readers may benefit from a brief explanation of the roles these genes play in EC proliferation and their significance in the context of MI.
10. Are there any notable genes or factors within the extrinsic apoptotic pathway that exhibited particularly significant changes in expression, and do these changes have known functional consequences?
11. Regarding the genes from the intrinsic pathway and those involved in both intrinsic and extrinsic pathways, what is the significance of their slight elevation in mRNA expression up to day 3 after coronary occlusion, and how might this relate to the progression of MI?
12. Could you discuss the potential implications of these findings for understanding the molecular mechanisms of EC apoptosis in MI and how they might be applied to clinical practice or future research?
13. Result sections 2.5 and 2.6 present a list of biological processes (BPs) related to angiogenesis regulation, but it doesn't provide a clear context for why these processes are relevant in the setting of MI and repeats the same information about the dynamics of angiogenesis-related BPs without adding substantial new insights. It could be more concise and focused.
14. Figures 6A and 6B may require more explanation to understand the data, especially considering the extensive number of genes involved.
15. Could authors provide some insights into how early changes in mRNA expression post-MI might correlate with the progression of the condition or microvascular injury resolution?
16. The discussion looks very lengthy and covers a wide range of topics related to EC biology, MI, angiogenesis, and gene expression. It might benefit from improved organization and structuring to follow the discussion more easily. The findings regarding EC apoptosis and proliferation, angiogenesis modulation, and the expression of pro/anti-angiogenic factors. While it presents the data comprehensively, there could be a more detailed interpretation and discussion of the implications of these findings. How do these gene expression changes relate to microvascular injury resolution and MI outcomes? While the study provides detailed insights into gene expression changes, it's essential to discuss the clinical significance of these findings. How might this knowledge be translated into clinical practice or potential therapeutic strategies for MI patients? Discuss any limitations of the study, such as the use of animal models or potential confounding factors, and their impact on the interpretation of the results. Highlight the novelty of the findings and suggest potential directions for future research. What are the key takeaways that advance our understanding of MI and microvascular injury? Provide more information about the functions of specific genes mentioned in the discussion, such as their roles in EC proliferation, apoptosis, or angiogenesis. How do these genes fit into the larger picture of MI pathophysiology?
17. In conclusion, discuss the briefing for additional experiments or studies that are needed to build upon the findings presented here.
18. The resolution of the figure is very poor.
Comments on the Quality of English Languagelittle improvement is required
Author Response
Reviewer #2: The authors aimed in this study to identify genes involved in post-myocardial infarction (MI) endothelial cell (EC) functions using RNA-sequencing data from mice subjected to coronary ischemia. The results show that apoptosis-related processes are active up to day 3 after ischemia, while EC proliferation begins from day 3 onwards. Apoptosis in ECs post-MI involves both extrinsic and intrinsic pathways, and there are changes in gene expression related to new vessel formation and angiogenesis regulation. This study provides valuable insights into the role of ECs after MI, potentially advancing our understanding of their significance. My questions and comments are:
- The authors repeated the information on microvascular injury and the importance of understanding its molecular pathways in the introduction. A more concise and direct presentation could make an impact on the introduction.
First, we would like to thank you for the exhaustive revision of our manuscript.
We fully agree that there is duplicity within the introduction regarding the information on post-myocardial infarction microvascular injury and the importance of understanding its molecular pathways. For that reason, Introduction section has been simplified and the relevance of further comprehension of the mechanisms underlied microvascular injury has been mentioned once:
Introduction section, page 2, paragraph 1:
“Since preliminary results suggesting that complete microvascular restoration after MI might be a realistic endpoint to attenuate left ventricular remodeling and ameliorate prognosis [8,9], unravelling the dynamics of the molecular pathways implicated in post-MI loss of EC viability and microvascular injury neoangiogenesis-mediated reparation is of great interest to better understand MI pathophysiology and reduce the occurrence of post-MI adverse events”.
- There are missing specific research gaps to explain the significance of microvascular injury and neo-angiogenesis regarding the study aims. Elaboration is required to address the hypothesis /objectives such as the specific genes or pathways the study intends to investigate.
This is a very appropriate suggestion since an in-depth comprehension of the state-of-art is mandatory to properly propose the hypothesis and objectives of the present study.
Briefly, abnormalities in myocardial microvasculature have been extensively addressed both in clinical and experimental settings. Concretely, the number of microvessels and the ultrastructure of the endothelial cell (ECs) suffer significant changes due to the ischemia-reperfusion process. For that reason, our first aim was to fully characterize the molecular key factors implicated in this process. Consequently, an in-depth analysis of the genes implicated in both ECs apoptosis and proliferation is of interest to better understand myocardial infarction (MI) pathophysiology and design novel therapeutic options aiming at preventing loss of ECs viability following MI.
Indeed, various investigations have concluded that angiogenesis is activated soon after ischemia onset to re-establish microvasculature injury. On top of that, a complete restore of microvascular density within the infarcted tissue is crucial to avoid a more compromised long-term cardiac function and prognosis. Taking this into account, our study also aims to characterize the factors underlying this process to establish the basis for designing further experiments aiming at broadening the understanding of post-MI angiogenesis activation and regulation.
To illustrate these two ideas, Introduction section has been re-written with the objective of highlighting the research gaps regarding the role of ECs in MI setting and, consequently, explaining the scientific relevance of our study.
Introduction section, page 1, paragraph 2 and page 2, paragraph 1 and 2:
“Massive disruption in endothelial cell (EC) lining has been reported after MI, which activates cell apoptosis and facilitates capillary obstruction and extravasation of blood content into the interstitium [1,3]. Indeed, severe reduction in microvessel density and abnormal ultrastructure in ECs from small capillaries occur few minutes after MI induction [4,5]. Contrarily, our own organism quickly release pro-angiogenic factors tending to repair microcirculation loss, as reflected in both in humans and in highly controlled models of MI [6,7]. From the very beginning of ischemia, factors with intense pro-angiogenic activity are increased in the patients’ plasma [4,8]. Furthermore, the peripheral blood of experiments and of patients with MI (from the onset of ischemia until several weeks afterwards) can exert a potent pro-angiogenic stimulus on harvested coronary endothelial cells [4]. This probably mediates the spontaneous regeneration of the microvasculature that, in general, successfully ends a few weeks after reperfusion. Since preliminary results suggesting that complete microvascular restoration after MI might be a realistic endpoint to attenuate left ventricular remodeling and ameliorate prognosis [8,9], unravelling the dynamics of the molecular pathways implicated in post-MI loss of EC viability and microvascular injury neoangiogenesis-mediated reparation is of great interest to better understand MI pathophysiology and reduce the occurrence of post-MI adverse events.
Over the last few decades, biomedical research has been revolutionized by the appearance of high-throughput sequencing technologies and several studies have been carried out in myocardium isolated from experimental models of MI [10-15]. A previous meta-analysis from our group using RNA-sequencing datasets was focused on identifying genes involved in post-MI extracellular matrix turnover and the specific collagen subunits and metalloproteinases participating in this scenario [16]. However, data specifically addressing EC viability due to the balance between apoptotic and proliferation pathways and key molecular factors regulating post-MI angiogenesis are scarce”.
- Could authors elaborate on the process of selecting differentially expressed genes (DEGs) and the methodology used for functional enrichment analysis with Gene Ontology terms? The methods sections need improvement. What statistical analysis was used for each experiment?
Based on the reviewer’s comment, a more detailed explanation regarding the process of selecting differential expressed genes and the methodology used for functional enrichment analysis with Gene Ontology terms was included in the revised version of the manuscript.
Methods section, page 16, paragraphs 3 and 4:
“Differential expression analysis was performed with DESeq2 (1.30.1) [44] to compare all the infarcted groups with controls. A p-adjusted value< 0.05 and a Log2FoldChange≥ 2 cutoff were used for determining DEGs. Briefly, DESeq2 normalizes and calculates the dispersion of the raw readings of each study group and carries out and performs a hypothesis using a Wald test.
Functional enrichment analysis of DEGs was performed with the Cluster Profiler package [45] using GO annotations [46] based on February 2023 data grouping the DEGs according to the BP category and adjusting the p-value with the Benjamin-Hochberg method. Using ClusterProfiler, the comparison of the BPs between different study groups was carried out (between mice with different ischemia times and control), obtaining functional profiles of each study group”.
- Were there any significant findings or trends observed in the functional enrichment analysis related to EC proliferation, apoptosis, and angiogenesis regulation?
After performing functional enrichment analysis, those GOs related to EC proliferation, apoptosis, and angiogenesis regulation were illustrated in Figure x. Concretely, the size of the circle indicates the number of overexpressed genes, whereas the colors ranging from red to blue represent adjusted p-value as shown in the legend. Indeed, in the revised version of the manuscript and thanks to Reviewer 2, Tables x and x were added to list the main genes related to EC proliferation and apoptosis together with their Log2FoldChange and the p-adjusted value.
In terms of EC proliferation and apoptosis, an increase in the expression of BPs related to ECs apoptosis, specifically Regulation of endothelial cell apoptotic process (GO: 200035) and Endothelial cell apoptotic process (GO: 0072577), was detected the first days after ischemia onset. Contrarily, significantly augmented levels of BPs involved in endothelium proliferation, namely Endothelial cell proliferation (GO: 0001935) and Regulation of endothelial cell proliferation (GO: 0001936), were observed from day 3 onwards. These data indicates that post-MI EC injury begins soon few hours after coronary occlusion, as reflected by the overrepresentation of key genes involved in EC apoptosis. However, mechanisms aiming at restoring EC viability are also trigger from day 3 onwards, probably with the objective of restablishing vascular homeostasis within the infarcted heart.
Regarding angiogenesis regulation, different pathways are activated in the MI scenario. Based on our meta-analysis, a clear enlargement at all ischemic times after coronary occlusion was detected in the following angiogenesis-related BPs: Regulation of angiogenesis (GO: 0045765) and Positive regulation of angiogenesis (GO: 0045766). The involvement of the Endothelial cell migration (GO: 0043542) and Blood vessel endothelial cell migration (GO: 0043534) BPs was reported within the infarct region at the sub-acute phase (7-14 days). Contrarily, negative regulation of angiogenesis started 3–7 days post-MI, as reflected by overexpression of the BPs Negative regulation of angiogenesis (GO: 0016525), Negative regulation of blood vessel morphogenesis (GO: 2000181), and Negative regulation of blood vessel diameter (GO: 0042310) (Figure 2). The activation of angiogenesis-related BPs few hours after ischemic insults indicates that an endogenous tendency to resotre microvascular density within the infarcted tissue is initiated very soon after MI induction, even when EC apoptosis is taking place. These results have been also detected at protein level since circulating levels of pro-angiogenic factors can be noticed few minutes following MI induction.
All these issues have been raised in the revised version of the manuscript:
Results section, page 3, paragraph 2:
“A clear augmentation in the expression of BPs related to ECs apoptosis, namely Regulation of endothelial cell apoptotic process (GO: 200035) and Endothelial cell apoptotic process (GO: 0072577), was noticed at initial stages (days 1–3) of ischemia. Contrarily, significantly increased levels of BPs implicated in endothelium proliferation, specifically Endothelial cell proliferation (GO: 0001935) and Regulation of endothelial cell proliferation (GO: 0001936), were distinguished from day 3 onwards (Figure 1)”.
Results section, page 8, paragraph 3 and page 9, paragraph 1:
“According to their dynamics, clear augmentation (with an adjusted p-value lower than 0.01) at all times after coronary occlusion was detected in the following angiogenesis-related BPs: Regulation of angiogenesis (GO: 0045765) and Positive regulation of angiogenesis (GO: 0045766). The involvement of the Endothelial cell migration (GO: 0043542) and Blood vessel endothelial cell migration (GO: 0043534) BPs was reported within the infarct region at the sub-acute phase (7-14 days). Contrarily, negative regulation of angiogenesis started 3–7 days post-MI, as reflected by overexpression of the BPs Negative regulation of angiogenesis (GO: 0016525), Negative regulation of blood vessel morphogenesis (GO: 2000181), and Negative regulation of blood vessel diameter (GO: 0042310) (Figure 1)”.
- Could the author elaborate on the significance of the bimodal behavior observed in the upregulated biological processes (BPs) related to endothelial cell (EC) proliferation and apoptosis after MI?
Based on our results, we have detected that BPs related to EC apoptosis are upregulated up to day 3 after ischemia onset, whilst those implicated in EC proliferation displayed an augmented expression from day 3 onwards.
Others and we have previously reported that microvascular ECs suffer significant damage soon after ischemia onset. Concretely, severe decline in microvessel density and abnormal ultrastructure in ECs from small capillaries (i.e. thinner and disrupted cytoplasm, chromatin condensation, or absence of transcytotic vesicles) occur few minutes after MI induction (Hollander MR, et al. 2016, Rios-Navarro C, et al. 2018). These data are in line with our novel results regarding the augmented mRNA levels of genes implicated in EC apoptosis up to day 3 post-MI.
Although microvascular density diminished after ischemic insult, clinical and experimental data pointed out that new vessel formation via angiogenesis activation actively participates in MI scenario. To do so, ECs proliferation, division and migration is crucial and, that is why genes implicated in EC proliferation may be overrepresented from day 3 onwards.
Taking all together, ECs suffer severe damage after coronary occlusion as reflected by a significant decay in microvascular density, abnormal capillary ultrastructure, and activation of genes implicated in EC apoptosis. Contrariwise, mechanisms involved in ECs proliferation and survivorship are also overrepresented from day 4 to chronic phases to restore the number of microvessels within the infarcted myocardium.
- Results are not well explained. What specific genes or factors were identified within the Regulation of endothelial cell apoptotic process and Endothelial cell apoptotic process biological processes during the initial stages of ischemia (days 1-3)?
To comprehensively understanding the key molecular factors implicated in post-MI EC apoptosis, Table 3 has been included in the revised version of the manuscript illustrating those genes overrepresented in the BPs Endothelial cell apoptotic process (GO: 0072577) and Regulation of endothelial cell apoptotic process (GO: 200035) during the initial stages of ischemia (day 1-3).
Results section, page 7, paragraph 3:
“Based on our results, serpine1 and thbs1 exhibited the highest transcriptomic changes at very acute phases following MI. Indeed, tnf, angptl4, tert, and il10 also displayed significant increase in mRNA levels few hours after ischemia onset (Table 3)”.
- The authors should provide more details about the genes or pathways that were associated with Endothelial cell proliferation and Regulation of endothelial cell proliferation from day 3 onwards in the results.
According to the reviewer’s suggestion, Table 2 has been added in order to illustrate the genes identified within the BPs Endothelial cell proliferation (GO: 0001935) and Regulation of endothelial cell proliferation (GO: 0001936) from day 3 to chronic phases together with the p-adjusted value and the Log2FoldChange.
Results section, page 13, paragraph 3:
“Concretely, hmox1 and arg1 displayed the highest transcriptomic expression 3 days after coronary occlusion, whereas mdk, scg2, gdf2, and tnmd showed strong changes in mRNA levels from day 7 onwards”.
- Interested to know whether any specific genes or pathways exhibited particularly strong changes in expression within these biological processes during the study.
We would like to thank the reviewer for these three suggestions about poinpinting which specific genes exhibited strong mRNA changes within the BPs related to EC apoptosis and proliferation. We definetely believe it increases the scientific relevance of our manuscript. As previously mentioned, Tables 2 and 3 listed the main genes related to EC proliferation and apoptosis together with Log2FoldChange and the p-adjusted value.
In terms of EC proliferation and its regulation, hmox1 and arg1 displayed the highest mRNA levels 3 days after coronary occlusion, whereas mdk, scg2, gdf2, and tnmd showed strong changes in their transcriptomic expression from day 7 to chronic phases.
Regarding genes participating in EC apoptosis and its regulation, serpine1 and thbs1 displayed the highest transcriptomic changes at very acute phases following MI. Indeed, tnf, angptl4, tert, and il10 are also key factors showing enlarged mRNA levels few hours after ischemia onset.
These issues have been raised in the revised version of the manuscript:
Results section, page 5, paragraph 2 and page 6, paragraph 1:
“Since increased EC proliferation was initiated at day 3 after coronary occlusion (Figure 1), Table 2 illustrates that 38 genes identified within the BPs Endothelial cell proliferation (GO: 0001935) and 36 included in Regulation of endothelial cell proliferation (GO: 0001936) exhibited enlarged mRNA expression. According to our data, hmox1 and arg1 displayed the highest mRNA 3 days after coronary occlusion, whereas mdk, scg2, gdf2, and tnmd showed strong changes in transcriptomic expression from day 7 to chronic phases”.
Results section, page 7, paragraph 3:
“First, Table 3 shows a total of 16 genes included in the BPs Endothelial cell apoptotic process (GO: 0072577) and Regulation of endothelial cell apoptotic process (GO: 200035), which displays an enlarged mRNA expression during the initial stages of ischemia (days 1-3). Based on our results, serpine1 and thbs1 exhibited the highest transcriptomic changes at very acute phases following MI. Indeed, tnf, angptl4, tert, and il10 also displayed significant increase in mRNA levels few hours after ischemia onset (Table 3)”.
- While the result section 2.3 reports changes in gene expression patterns, it doesn't delve into the potential functional implications of these changes. Readers may benefit from a brief explanation of the roles these genes play in EC proliferation and their significance in the context of MI.
According to our results, the mRNA levels of up to 38 genes involved in regulation of EC proliferation are significantly enlarged. Concretely, hmox1 and arg1 displayed the highest transcriptomic expression 3 days after coronary occlusion, whereas mdk, scg2, gdf2, and tnmd showed strong changes in mRNA levels from day 7 onwards. To delve into the potential functional implications of the overrepresentation of these genes, further information about their role in MI context has been reported:
Heme oxygenase 1 (hmox1) displays antioxidant and antiapoptotic effects in response to certain stresses, including ischemia-reperfusion injury. In MI scenario, heme oxygenase has been demonstrated to reduce infarct size and left ventricular remodelling in murine models of permanent ischemia (Kusmic C, et al. 2014) and exert cardioprotective effects via attenuating cardiomyocyte senescence (Shan H, et al. 2019).
Arginase 1 (arg1) participates in the urea cycle by converting arginine to ornithine and urea. Circulating levels of arginase 1 are increased in MI patients compared to controls. Indeed, a positive correlation between the amount of arginase 1 and gensini score was also detected (Zhang R, et al. 2020). Although its role in MI scenario is not yet explored, arginase 1 could be a potential biomarker for MI patients.
Midkine (mdk), a heparin-binding growth factor, participtes in inflammation by inducing leukocyte infiltration, chemokine expression and suprresion of regulatory T cells expansion. Specifically in MI scenario, administration of midkine in rats submitted to MI prevents left ventricular remodelling by promoting angiogenesis and reducing collagen deposition and apoptosis activation (Sumida A, et al. 2009).
Growth differentiation factor 2 (gdf2), also known as bone morphogenetic protein 9, is a member of the transforming growth factor superfamily of proteins. A recent study reported that loss of bone morphogenetic protein 9 promotes cardiac fibrosis, collagen degradation, and cardiac rupture after MI, probably by increasing the expression of metalloproteinase 9 (Bhave S, et al. 2023).
Secretogranin II (scg 2) and Tenomodulin (tnmd) has been barely explored in the setting of MI. Consequently, further investigation is required to clarify its participation after ischemia-reperfusion injury.
To sum up, more information about the potential role of the highly expressed genes related to EC proliferation has been added in the revised version of the manuscript.
Discussion section, page 13, paragraphs 3 and 4:
“According to Table 2, the mRNA levels of up to 38 genes involved in regulation of EC proliferation are significantly enlarged. Concretely, hmox1 and arg1 displayed the highest transcriptomic expression 3 days after coronary occlusion, whereas mdk, scg2, gdf2, and tnmd showed strong changes in mRNA levels from day 7 onwards. In the MI scenario, heme oxygenase (hmox 1) has been demonstrated to reduce infarct size and left ventricular remodeling in murine models of permanent ischemia [19] and exert cardioprotective effects via attenuating cardiomyocyte senescence [20], whereas midkine (mdk) administration prevents left ventricular remodeling by promoting angiogenesis and reducing collagen deposition and apoptosis activation [21]. Indeed, the loss of growth differentiation factor 2 (gdf2) promotes cardiac fibrosis, collagen degradation, and cardiac rupture after MI via metalloproteinase 9 [22]. Lastly, circulating levels of arginase 1 (arg1) are augmented in MI patients compared to controls and positively correlate with gensini score [23]. Contrariwise, from our knowledge, studies regarding the participation of secretogranin II (scg2) and tenomodulin (tnmd) in MI scenario are scarce.
Vascular endothelial growth factor actively participates in EC proliferation to spontaneously promote post-MI microvascular injury resolution via angiogenesis activation [8]. Our data revealed the overrepresentation of several positive regulators of EC proliferation (i.e. scg2, ccr3, and vash2) (Figure 2), which have been linked to the vascular endothelial growth factor signaling pathway in other disease entities [24,25]. This highlights the utility of further investigating their specific involvement in the pathophysiology of MI as promising biomarkers and/or novel therapeutic options”.
- Are there any notable genes or factors within the extrinsic apoptotic pathway that exhibited particularly significant changes in expression, and do these changes have known functional consequences?
According to our results, a total of 13 genes related to the extrinsic signaling pathway displayed changes in their transcriptomic expression after ischemia onset. Of them, thbs1, serpine1, scg2, and tert presented the higher mRNA levels from day 1 onwards.
Results section, page 7, paragraph 4:
“Of these, thbs1, serpine1, scg2, and tert presented a higher Log2FoldChange compared to fasl, bmp4, icam1, and tnfaip3. In contrast, the transcriptomic levels of il4, gepr1, and foxo3 were downregulated very soon after coronary occlusion (6–24 hours) and the gene expression of fga remained unaltered (Figure 3)”.
To delve into the potential functional implications of the overrepresentation of these genes, further information about their role in MI context has been reported:
Thrombospondin 1 (thbs1) is rapidly activated in response to different stresses, such as myocardial ischemia. Concretely thrombospondin-1 has been reported to participate in MI context via different mechanisms, including inhibition of inflammation by promoting neutrophils apoptosis, myofibroblast differentation, acceleration of fibrotic scar formation, and activation of apoptosis in ECs at the microvascular level (Kirk JA, et al. 2016).
Serpine 1 (serpine 1), also known as plasminogen activator inhibitor-1, has been associated with higher 5-year risk of death in ST-segment elevation MI patients (Pavlov M, et al. 2018) and ventricular disfunction (Shimizu T, et al. 2016). In experimental models, the presence of plasminogen activator inhibitor-1 was increased probably via transforming growth factor-β and tumor necrosis factor-α and participats in cardiac fibrosis and ventricular remodeling (Takeshita K, et al. 2004).
Secretogranin II (scg 2), however, has been barely explored in the setting of MI. From our knowledge, secretogranin II is a potential biomarker for heart failure development. Contrarily, studies evaluating its implication after MI are limited. For that reason, further investigation is required to clarify its participation after ischemia-reperfusion injury.
Telomerase reverse transcriptase (tert) has been extensively studied after MI. Concretely, telomerase reverse transcriptase overrepresentation was associated with reduced infarct size and better systolic function in murine models of permanente ischemia (Bar C, et al. 2014) and a decay in cardiomyocyte apoptosis (Oh H, et al. 2009). In this line, the lack of telomerase reverse transcriptase correlates heart failure development (Ait-Aissa K, et al. 2019).
To sum up, more information about the potential role of the highly expressed genes related to extrinsic apoptosis has been added in the revised version of the manuscript.
Discussion section, page 14, paragraph 4:
“Indeed, thbs1, serpine1, scg2, and tert regulate the extrinsic apoptotic pathway and presented the higher mRNA levels from day 1 onwards. Specifically, thrombospondin 1 (thbs1) participates in MI context via inhibition of inflammation by promoting neutrophils apoptosis, myofibroblast differentiation, acceleration of fibrotic scar formation, and activation of apoptosis in ECs at the microvascular level [31], whereas telomerase reverse transcriptase (tert) overrepresentation correlates with reduced infarct size and better systolic function in murine models of permanent ischemia [32] and a decay in cardiomyocyte apoptosis [33]. Regarding serpine 1, increased circulating levels are associated with higher 5-year risk of death in STEMI [34] and ventricular disfunction [35]. In experimental models, the presence of serpine 1 is probably enlarged via transforming growth factor-β and tumor necrosis factor-α and participates in cardiac fibrosis and ventricular remodeling [36]”.
- Regarding the genes from the intrinsic pathway and those involved in both intrinsic and extrinsic pathways, what is the significance of their slight elevation in mRNA expression up to day 3 after coronary occlusion, and how might this relate to the progression of MI?
This is a very interesting question. The mechanism of apoptosis mainly consists of two core pathways involved in inducing apoptosis: extrinsic pathway and intrinsic pathway.
Extrinsic pathway is activated when extracellular ligands such as tumor necrosis factor, Fas ligand, and TNF-related apoptosis-inducing ligand interact with the extracellular domain of the death receptors. Circulating levels of TNF and Fas ligand are augmented few hours after ischemia onset due to the ischemia-induced necrosis on cardiomyocytes. This issue could explain our results regarding the upregulation of genes involved in extrinsic apoptosis soon after MI induction.
The intrinsic pathway, however, refers to mainly mitochondrial-mediated apoptotic pathway and is triggered by various intracellular stresses, including oxidative stress. In the MI scenario, oxidative stress is reported to occur the first hours after MI because of the arrival of oxygenated blood into the ischemic myocardium. Consequently, genes implicated in the intrinsic apoptotic process are only overrepresented up to day 3 after coronary occlusion.
Although the activation of apoptosis on EC following MI has been previously reported, data regarding the concrete apoptotic pathway implicated in this process has been barely explored. Therefore, our data could shed light into the mechanisms underlying post-MI EC apoptosis.
Results section, page 14, paragraphs 2, 3, 4, and 5:
“Furthermore, we also shed light on the specific apoptotic pathway (extrinsic or extrinsic) activated on ECs post-MI. Expanding on a previous study performed by single cell RNA sequencing in ECs submitted to permanent myocardial ischemia demonstrated the implication of the intrinsic signaling pathway via bax and trp53 upregulation [18], our data showed heightened mRNA expression of genes involved in intrinsic (n=2) and extrinsic (n=13) signaling pathways (Figure 3).
The extrinsic apoptotic pathway is triggered once ligands (i.e., tumor necrosis factor receptor and Fas) interact with their corresponding death receptors on the cell surface [28]. In this scenario, tnfaip3, strongly activated by tumor necrosis factor and interleukin-1 [29], displays higher transcriptomic levels on the infarcted myocardium throughout the entire ischemic process compared to control myocardium. A recent study reported a positive correlation between TNFAIP3 protein concentration on circulating monocytes from MI patients and high-sensitivity troponin T levels [30].
Indeed, thbs1, serpine1, scg2, and tert regulate the extrinsic apoptotic pathway and presented the higher mRNA levels from day 1 onwards. Specifically, thrombospondin 1 (thbs1) participates in MI context via inhibition of inflammation by promoting neutrophils apoptosis, myofibroblast differentiation, acceleration of fibrotic scar formation, and activation of apoptosis in ECs at the microvascular level [31], whereas telomerase reverse transcriptase (tert) overrepresentation correlates with reduced infarct size and better systolic function in murine models of permanent ischemia [32] and a decay in cardiomyocyte apoptosis [33]. Regarding serpine 1, increased circulating levels are associated with higher 5-year risk of death in STEMI [34] and ventricular disfunction [35]. In experimental models, the presence of serpine 1 is probably enlarged via transforming growth factor-β and tumor necrosis factor-α and participates in cardiac fibrosis and ventricular remodeling [36].
The intrinsic pathway, meanwhile, is triggered via the endoplasmic reticulum and mitochondria [28]. Our findings point to an overrepresentation of xbp1, a transcription factor that modulates not only the cellular response to endoplasmic reticulum stress but also vascular endothelial growth factor-induced angiogenesis in adult tissues submitted to ischemia [37]. Despite its role in the mechanisms underlying various cardiovascular diseases including cardiac hypertrophy and heart failure [37], few studies have addressed the involvement of xbp1 in the MI context”.
- Could you discuss the potential implications of these findings for understanding the molecular mechanisms of EC apoptosis in MI and how they might be applied to clinical practice or future research?
Ischemia-reperfusion insult has traditionally been linked to necrosis and apoptosis in cardiomyocytes. However, studies evaluating the occurrence of apoptosis are microvascular EC are scarce.
In vitro and in vivo experiments have shown that endothelial cells resisted prolonged periods (24-48 hours) of hypoxia by triggering protective mechanisms (i.e., increased levels of malondialdehyde, superoxide dismutase, and hypoxia-inducible factor-1A). However, a previous article in ST-segment elevation MI patients suggested that their serum isolated after coronary revascularization induced apoptosis in endothelial cells in vitro (Rios-Navarro C, et al. 2023). In this line, a recent study performed in 153 patients with acute coronary syndrome found an increased number of endothelial-derived microvesicles related to apoptosis, especially in those displaying STEMI (Zacharia N, et al. 2020). Therefore, scrutinizing which molecular mechanism of EC apoptosis are triggered in MI will shed light into the role of endothelium on this scenario.
On top of that, a previous study performed by single cell RNA sequencing in ECs submitted to permanent myocardial ischemia demonstrated the implication of the intrinsic signaling pathway via bax and trp53 upregulation (Tombor LS, et al. 2021). However, as far as we are concern, our data firstly showed heightened mRNA expression of genes involved in both intrinsic (n=2) and extrinsic (n=13) apoptotic signaling pathways.
Our study aims to exhibit potential candidates to regulate EC apoptosis in the context of MI. However, we are aware that these results need further validation not only at protein level in experimental models but also at bedside. One approach could be to determine circulating levels of these factors involved in post-MI EC apoptosis and afterwards, correlate them with cardiovascular magnetic resonance-derived indices (i.e. extension of microvascular obstruction, edema, and intramyocardial hemorrhage). Additionally, therapeutic options aiming at minimizing EC apoptosis after ischemia/reperfusion injury would be also of great interest. Unfortunately, these experiments are out of the scope of our study.
Discussion section, page 14, paragraph 6:
“our study reveals new insights into molecular targets probably implicated in EC apoptosis regulation, but that have barely been scrutinized on MI context so far. These novel data could establish the basis for designing further experiments aiming at broadening the understanding of EC participation post-MI as well as setting up new potential circulating biomarkers of compromised resultant cardiac structure or therapeutic options to dowregulate EC apoptosis after ischemia/reperfusion injury”.
- Result sections 2.5 and 2.6 present a list of biological processes (BPs) related to angiogenesis regulation, but it doesn't provide a clear context for why these processes are relevant in the setting of MI and repeats the same information about the dynamics of angiogenesis-related BPs without adding substantial new insights. It could be more concise and focused.
Following the reviewer suggestion, an in-depth explanation of the relevance of these processes in the context of myocardial infarction has been added:
Results section, page 9, paragraph 2
“This indicates that new vessel formation is activated soon after ischemia onset aiming at restoring microvascular density within the infarcted tissue, whereas negative regulators of angiogenesis probably are upregulated few days (3 to 7) post-MI. Another mechanism to promote the formation of new vessels into the ischemic heart is the migration of ECs, whose genes are mainly overrepresented from day 7 onwards”.
Results section, page 10, paragraph 1:
“To sum up, Figure 4 represents potential candidates to regulate angiogenesis in MI context. Concretely, positive regulators of angiogenesis are activated at all ischemic times after coronary occlusion, representing a more than twofold number of genes compared to negative regulators aiming at promoting the formation of new vessels within the ischemic heart”.
- Figures 6A and 6B may require more explanation to understand the data, especially considering the extensive number of genes involved.
To expand the description regarding the transcriptomic expression of pro/anti-angiogenic factors, some changes, including the name of specific genes, have been introduced.
Results section, page 10, paragraph 2 and page 11, paragraph 1:
“Based on their dynamics, seven genes (csf3, cd44, itgb3, tnf, sema3f, kit, and tgfb1) showed augmented mRNA levels very soon after coronary occlusion and this trend was sustained at acute phase (3–7 days). A second group was made up of 18 genes (ptn, fst, fn1, hgf, and ephb2, among others) displaying elevated mRNA expression from day 7 to chronic phases, and lastly, 26 genes encoding pro-angiogenic proteins (ceacam1, flt1, prox1, and fgf1, among others) showed reduced or slightly augmented levels. Indeed, the genes with the highest mRNA levels were csf3 (at initial phase) and ptn (at late phase) (Figure 5A)”.
Results section, page 11, paragraph 2:
“Of note, chga displayed heightened mRNA levels in all MI groups compared to sham, whereas 11 different genes, including angpt2, angpt4, cxcl10, and tnfsl15, were overrepresented from day 7 onwards. Contrariwise, three genes (vegfb, pgk, and tie1) exhibited diminished mRNA expression and the transcriptomic levels of 13 genes encoding anti-angiogenic proteins (i.e. timp2, angptl1, timp3, and adamts1) displayed mild augmentation (Figure 5B)”.
- Could authors provide some insights into how early changes in mRNA expression post-MI might correlate with the progression of the condition or microvascular injury resolution?
Based on the preliminary clinical results, the presence of persistent microvascular injury in chronic phases following MI may be related to a more compromised long-term cardiac function and prognosis. From the very beginning of ischemia, products with intense pro-angiogenic activity are increased in the patients’ plasma. This probably mediates the spontaneous regeneration of the microvasculature that, in general, successfully ends a few weeks after reperfusion. Both in experimental porcine models (with sequential assessment of microcirculation by intracoronary injection) and in patients with reperfused ST-segment elevation MI (sequentially studied with intracoronary contrast echocardiography and with cardiovascular magnetic resonance) we have confirmed this spontaneous and salutary trend.
Taking this as an initial idea, we are determined to further clarify the molecular mechanisms underlied the dynamics of microvascular ECs following MI. First, we scrutinized which genes participate in ECs apoptosis, which may be ultimately responsible for the massive decay in microvessel density detected few minutes after ischemia onset. Second, exploring key factors regulating angiogenesis and EC proliferation is of relevance to further understand the mechanisms implicated in the spontaneous tendency towards recovery the number of microcapillary within the infarcted tissue.
This issue has been briefly explained in Introduction section, page 1, paragraph 2 and page 2, paragraph 1:
“Massive disruption in endothelial cell (EC) lining has been reported after MI, which activates cell apoptosis and facilitates capillary obstruction and extravasation of blood content into the interstitium [1,3]. Indeed, severe reduction in microvessel density and abnormal ultrastructure in ECs from small capillaries occur few minutes after MI induction [4,5]. Contrarily, our own organism quickly release pro-angiogenic factors tending to repair microcirculation loss, as reflected in both in humans and in highly controlled models of MI [6,7]. From the very beginning of ischemia, factors with intense pro-angiogenic activity are increased in the patients’ plasma [4,8]. Furthermore, the peripheral blood of experiments and of patients with MI (from the onset of ischemia until several weeks afterwards) can exert a potent pro-angiogenic stimulus on harvested coronary endothelial cells [4]. This probably mediates the spontaneous regeneration of the microvasculature that, in general, successfully ends a few weeks after reperfusion. Since preliminary results suggesting that complete microvascular restoration after MI might be a realistic endpoint to attenuate left ventricular remodeling and ameliorate prognosis [8,9], unravelling the dynamics of the molecular pathways implicated in post-MI loss of EC viability and microvascular injury neoangiogenesis-mediated reparation is of great interest to better understand MI pathophysiology and reduce the occurrence of post-MI adverse events”.
- The discussion looks very lengthy and covers a wide range of topics related to EC biology, MI, angiogenesis, and gene expression. It might benefit from improved organization and structuring to follow the discussion more easily. The findings regarding EC apoptosis and proliferation, angiogenesis modulation, and the expression of pro/anti-angiogenic factors. While it presents the data comprehensively, there could be a more detailed interpretation and discussion of the implications of these findings. How do these gene expression changes relate to microvascular injury resolution and MI outcomes? While the study provides detailed insights into gene expression changes, it's essential to discuss the clinical significance of these findings. How might this knowledge be translated into clinical practice or potential therapeutic strategies for MI patients? Discuss any limitations of the study, such as the use of animal models or potential confounding factors, and their impact on the interpretation of the results. Highlight the novelty of the findings and suggest potential directions for future research. What are the key takeaways that advance our understanding of MI and microvascular injury? Provide more information about the functions of specific genes mentioned in the discussion, such as their roles in EC proliferation, apoptosis, or angiogenesis. How do these genes fit into the larger picture of MI pathophysiology?
We fully agree with the reviewer’s suggestion and consequently, Discusion section was re-written to shorten it and focus on the main findings of the present study.
First, we have included more information about the potential role of the highly expressed genes related to EC proliferation has been added in Discussion section, page 13, paragraph 3:
“According to Table 2, the mRNA levels of up to 38 genes involved in regulation of EC proliferation are significantly enlarged. Concretely, hmox1 and arg1 displayed the highest transcriptomic expression 3 days after coronary occlusion, whereas mdk, scg2, gdf2, and tnmd showed strong changes in mRNA levels from day 7 onwards. In the MI scenario, heme oxygenase (hmox 1) has been demonstrated to reduce infarct size and left ventricular remodeling in murine models of permanent ischemia [19] and exert cardioprotective effects via attenuating cardiomyocyte senescence [20], whereas midkine (mdk) administration prevents left ventricular remodeling by promoting angiogenesis and reducing collagen deposition and apoptosis activation [21]. Indeed, the loss of growth differentiation factor 2 (gdf2) promotes cardiac fibrosis, collagen degradation, and cardiac rupture after MI via metalloproteinase 9 [22]. Lastly, circulating levels of arginase 1 (arg1) are augmented in MI patients compared to controls and positively correlate with gensini score [23]. Contrariwise, from our knowledge, studies regarding the participation of secretogranin II (scg2) and tenomodulin (tnmd) in MI scenario are scarce”.
Second, we have described the potential role of the main genes participating in the extrinsic apoptotic pathway has been included in Discussion section, page x, paragraphs x and x:
Discussion section, page 14, paragraph 3:
“Indeed, thbs1, serpine1, scg2, and tert regulate the extrinsic apoptotic pathway and presented the higher mRNA levels from day 1 onwards. Specifically, thrombospondin 1 (thbs1) participates in MI context via inhibition of inflammation by promoting neutrophils apoptosis, myofibroblast differentiation, acceleration of fibrotic scar formation, and activation of apoptosis in ECs at the microvascular level [31], whereas telomerase reverse transcriptase (tert) overrepresentation correlates with reduced infarct size and better systolic function in murine models of permanent ischemia [32] and a decay in cardiomyocyte apoptosis [33]. Regarding serpine 1, increased circulating levels are associated with higher 5-year risk of death in STEMI [34] and ventricular disfunction [35]. In experimental models, the presence of serpine 1 is probably enlarged via transforming growth factor-β and tumor necrosis factor-α and participates in cardiac fibrosis and ventricular remodeling [36]”.
Next, we have also included some limitations of this study:
Discusion section, page 15, paragraphs 6 and 7:
“3.4. Limitations of the study
In our study, we are unable to detect DEGs among different datasets due to the statistical analysis performed.
These results were obtained at gene level and in mice submitted to permanent coronary ischemia; therefore, further validation at protein level and in samples from MI patients is necessary”.
Lastly, potential implications of these results have been already mentioned in the revised version of the manuscript:
Discussion section, page 14, paragraph 5:
“Collectedly, our study reveals new insights into molecular targets probably implicated in EC apoptosis regulation, but that have barely been scrutinized on MI context so far. These novel data could establish the basis for designing further experiments aiming at broadening the understanding of EC participation post-MI as well as setting up new potential circulating biomarkers of compromised resultant cardiac structure or therapeutic options to dowregulate EC apoptosis after ischemia/reperfusion injury”.
Discussion section, page 15, paragraph 5:
“Finally, it should be borne in mind that augmented gene expression does not always translate to the protein level. For that reason, next research step should be to corroborate these results at protein level both in experimental models of myocardial ischemia and, more important, in clinical scenario. However, our study proposes several key factors, which potentially participates in MI pathophysiology and could help establish the basis for designing further clinical and experimental studies addressed at broadening the range of potential angiogenesis regulators participating in post-MI microvascular repair”.
- In conclusion, discuss the briefing for additional experiments or studies that are needed to build upon the findings presented here.
This is a very interesting question because we believe this study is a platform for pointing out new potential candidates participating in MI pathophysiology. However, we are also aware that these results were obtained in experimental mice models and demonstrated at gene level. For that reason, next step should be to corroborate these results at protein level using different techniques such as proteomics, immunohistochemistry, and western blot. Concretely, we are now planning to test some potential molecular pathways at protein level in myocardial samples from both murine and swine models of reperfused and non-reperfused MI.
On top of that, the translation of these data to bedside, although complicate, would be of great interest for next investigations.
In the revised version of the manuscript, the potential application of these results into future research approaches are specified in Discussion section, page 15, paragraph 5:
“Finally, it should be borne in mind that augmented gene expression does not always translate to the protein level. For that reason, next research step should be to corroborate these results at protein level both in experimental models of myocardial ischemia and, more important, in clinical scenario. However, our study proposes several key factors, which potentially participates in MI pathophysiology and could help establish the basis for designing further clinical and experimental studies addressed at broadening the range of potential angiogenesis regulators participating in post-MI microvascular repair”.
- The resolution of the figure is very poor.
According to this suggestion, Figures were changes in the revised version of the manuscript.